# Outcomes of Patients with Lower Limb Loss after Using a Training Prosthesis: A Retrospective Case Series Study

**DOI:** 10.3390/healthcare12050567

**Published:** 2024-02-29

**Authors:** Doriane Pelzer, Charlotte Beaudart, Stephen Bornheim, Benoît Maertens de Noordhout, Cédric Schwartz, Jean-François Kaux

**Affiliations:** 1Physical Medicine, Rehabilitation and Sports Traumatology Department, University and University Hospital of Liège, 4000 Liège, Belgium; dpelzer@chuliege.be (D.P.); stephen.bornheim@uliege.be (S.B.); bmaertens@chuliege.be (B.M.d.N.); cedric.schwartz@uliege.be (C.S.); 2Clinical Pharmacology and Toxicology Research Unit, Department of Biomedical Sciences, Namur Research Institute for Life Sciences (NARILIS), Faculty of Medicine, University of Namur, 5000 Namur, Belgium; charlotte.beaudart@unamur.be

**Keywords:** training prosthesis, rehabilitation, amputation, outcomes

## Abstract

The aim of this retrospective case series study was to investigate outcomes in patients with lower limb loss based on whether or not they used a training prosthesis (TP) during rehabilitation. The medical records of 171 consecutive patients admitted to rehabilitation hospitalization between January 2014 and December 2018 following a major amputation of the lower limb were reviewed. Patients were categorized into two groups: patients who underwent rehabilitation with a TP and patients who did not use a TP. Outcomes (i.e., discharge destination, length of stay, number of sockets required, and number of the size adaptation of each socket, as well as functional level) were compared between groups. Of the 171 patients, 126 underwent rehabilitation with a TP, and 45 patients underwent rehabilitation without any TP. In conclusion, we found that patients who used a TP had a significantly shorter hospital length of stay when compared to those who did not. This length of stay for patients with TP was not influenced by age but was lowered by a higher body mass index (BMI), tibial instead of femoral amputation, and the male gender. No association was found between the use of TP and discharge destination, functional level, number of socket modifications, and number of sockets required.

## 1. Introduction

### 1.1. Background/Rationale

The leading causes of amputation worldwide are vascular diseases, followed by trauma, cancer, and finally congenital impairment [1,2]. Vascular causes are expected to remain the primary reason for amputation for the foreseeable future considering the prevalence of peripheral vascular disease and diabetes. In fact, between 2000 and 2010, the number of people with peripheral arterial disease increased by almost 25% worldwide and concerned high-, middle-, and low-income countries [3]. The global prevalence of diabetes in 2019 has been estimated at 9.3%, and projections for 2030 and 2045 predict an estimated prevalence of 10.2% and 10.9%, respectively [4].

Limb amputation remains a major cause of public health concern today [5], and amputee rehabilitation holds an important place and represents a major challenge for rehabilitation physicians [6,7]. Multicomponent exercise programs showed gait improvements during the early and chronic phases of recovery [8]. Esquenazi and Meier [6] proposed a nine-phase description of amputee rehabilitation: pre-operative, amputation surgery/reconstruction surgery, acute post-surgical, pre-prosthetic, prosthetic prescription, prosthetic training, community integration, vocational rehabilitation, and follow-up.

However, there is a lack of consensus regarding post-operative management strategies in amputee rehabilitation [9]. The choice of soft, rigid, or semi-rigid materials for post-operative dressing is substantially discussed in the literature. The superiority of the rigid system is often mentioned, and it is more favorable than soft systems in reducing stump volume, protecting it from external impacts, improving healing, or even reducing the time between amputation and the acquisition of prosthesis [10,11]. However, soft dressing is conventionally chosen, and this is most likely due to the fragility of the skin and because of the risk of stump injury, particularly vascular stumps, and the difficulties in producing a rigid system [12]. Finally, a recent Cochrane systematic review [13] concludes that there is no certainty on the superiority of rigid dressing compared to soft dressing in terms of stump healing, pain, skin side effects, the length of stay in hospital, and the time between amputation and prosthesis prescription. Interestingly, there seems to be no study on the influence of the type of dressing on comfort, quality of life, or financial aspects [13].

Regarding prosthetic rehabilitation, the second half of the 20th century saw the emergence of immediate post-operative prostheses (IPOPs) [12]. Evidence-based data on their efficacy are, however, still lacking in the literature [14,15]. However, the use of these prostheses seems to be associated with a decrease in the number of falls and surgical revisions [16], as well as a decrease in the time between amputation and the prescription of the personalized prosthesis [17]. Immediate prostheses are only prescribed in around 5% of cases, notably due to technical difficulties, skin fragility, and more difficult wound monitoring [18].

While IPOPs are used in the post-operative phase, it is possible to use another type of prosthesis during the pre-prosthetic phase of rehabilitation for training purposes. This training prosthesis (TP) comprises classic modular prosthetic elements fixed to a socket made of plaster and manufactured by following the usual stages of molding the stump and working a positive mold. Although its use is not uncommon, there is a lack of studies on this topic in the literature. However, its use could have an impact on several key rehabilitation outcomes [19], such as length of stay, discharge destination, quality of life, functionality, economic burden, prosthesis fitting, etc.

### 1.2. Objectives

The aim of our work is therefore to investigate different health rehabilitation outcomes after the rehabilitation of lower limb loss patients, and this is based on whether the patients used a training prosthesis during the rehabilitation program.

## 2. Methods

### 2.1. Study Design

This is a retrospective case–control study. It is organized according to STROBE recommendations.

### 2.2. Setting

The medical records of 208 consecutive patients admitted to rehabilitation centers between January 2014 and December 2018 following a major amputation of the lower limb were retrospectively reviewed. The two hospitals concerned were the University Hospital of Liège Site “CNRF” (Centre Neurologique de Réadaptation Fonctionnelle of Fraiture, Belgium) and the Regional Hospital Center of the Citadelle Site “Château Rouge” (Liège, Belgium). The study was conducted in accordance with the Declaration of Helsinki, and the protocol was approved by the respective Ethics Committees of both centers (2018—12-03V1; JL/bl/TDD2019/11—B412201940866).

### 2.3. Population

Patients were included in the study if they had undergone a major unilateral or bilateral amputation of the lower limb (above the ankle and below the hip), without restriction of age, comorbidity, or etiology. Major amputations were selected because they show a difference compared to minor amputations in terms of rehabilitation outcome (walking ability, quality of life, and dependency status) [20].

As the study deals with the consequences of using or not using a training prosthesis, patients were excluded from the study if there was no initial fitting intention for medical reasons or because of the patient’s choice.

All patients received 2 h of rehabilitation each day. The treatment included physiotherapy and occupational therapy to work on strengthening, gait rehabilitation, balance, and functionality in daily living activities. When necessary, they also received speech therapy, psychology, and neuropsychology care.

### 2.4. Variables

A training prosthesis is a rehabilitation tool used during the rehabilitation phase when the condition of the stump allows for it, mainly meaning that there was sufficient healing and the pain was bearable. A metal rod (with or without the knee articulated according to the femoral or tibial status of the amputation) was placed in the plaster cast at its upper end, and it was connected distally to a prosthetic foot. Pictures of the training prosthesis are available in Figure 1.

This prosthesis is not systematically used in the rehabilitation of the amputee patient because it requires that the patient portrays certain faculties and technical skills. The TP was made by members of the rehabilitation team who have undergone specialized training. The two rehabilitation centers included in this study were selected as both are in the same geographic area; the patients are usually treated by the same surgical teams, and the casting and prosthetics are made by the same teams of prosthetists. In addition, the rehabilitation centers followed the same rehabilitation protocols, and their objectives are usually similar. However, CHU-CNRF regularly integrates the training prosthesis technique into the rehabilitation of amputee patients, while CHR-Château Rouge does not use it.

The impact of TP was assessed based on several outcomes.

*Discharge destination*: Discharge destination concerns whether patients were allowed to go home or to a nursing facility when they were discharged. This orientation is considered to reflect the autonomy of the patient. The decision was made based on whether the patient required the implementation of adequate aids or not and if the aids provided sufficient autonomy at home, while living in an NF meant that patients were monitored and that there was constant help from qualified personnel if needed due to insufficient autonomy.

*Length of stay* in rehabilitation hospitalization is measured based on the number of days between admission to the rehabilitation center and discharge from the center with either continuing rehabilitation as an outpatient or stopping multidisciplinary rehabilitation with or without physiotherapy at home. The day of discharge from rehabilitative hospitalization represents the day on which no further benefit was expected from hospitalization.

*The number of sockets required and the number of size adaptations of each socket* are related to the stability of the evolution of the stump volume, and it also represents a potential financial factor. This point relates to the first prosthesis prescribed: the “evaluation” prosthesis (EP).

*Functional group*: At least six months after the prescription of the EP, the definitive prosthesis (DP) can be prescribed based on the category corresponding to the functional level of the patient. There are five categories (in accordance with the Belgian health insurance https://www.inami.fgov.be (accessed on 26 February 2024), which is also available in the Appendix B). Category 1 includes patients for whom no walking prospects are expected. The prosthesis fulfills a purely aesthetic function in this category. Category 2 covers patients with a greatly reduced walking function, and prostheses in this category essentially fulfill a transfer function. In category 3, patients have a reduced walking function but can move around without the help of a third person, provided that technical aids are available (walking frame, cane, etc.). The following two categories include patients who can move without technical aid. A walking test is performed for these patients based on their level of amputation. If the results of this test reach or exceed the required value, the patient is classified in category 5. Otherwise, they are classified in category 4. For this study, an additional category was also created, “category 0”, which includes patients who have not had a DP prescription, and this is most often because of prosthesis abandonment or because of very limited use or no desire to change the EP.

## 3. Bias

The sample size was not equivalent between the two groups. In order to test the robustness of the results, we carried out the same analysis on a new sample comprising 45 patients who did not receive a TP and 45 matched patients who received a TP. They were matched for gender and age (±5 years).

## 4. Study Size

Of all records obtained after the database review from 2014 to 2018, we excluded patients for whom there was no fitting intention. Afterward, we excluded patients with insufficient data for each outcome.

## 5. Quantitative Variables and Statistical Methods

The normality of continuous variables was checked by examining the histogram, the quantile–quantile plot, the Shapiro–Wilk test, and the difference between the mean and median values. Distribution was considered not normal if the data met less than 3 of the 4 conditions. Quantitative variables following a Gaussian distribution were expressed as the mean and standard deviation. Quantitative variables not following a Gaussian distribution were expressed as the median (25th percentile–75th percentile). Qualitative variables were described by absolute (n) and relative (%) frequencies. First, the samples of the two study groups (with TP versus without TP) were compared with each other to check if the characteristics of the population were represented in a similar manner between both groups. Secondly, rehabilitation outcomes were compared between groups. Finally, we performed subgroup analyses on gender, age (<65 years vs. ≥65 years), body mass index (BMI) (<25 kg/m^2^ vs. ≥25 kg/m^2^), and amputation level (unilateral femoral vs. tibial), and we measured possible associations between TP and outcomes within each of these subgroups. Because of the non-Gaussian distribution, the non-parametric Mann–Whitney U test was used to compare continuous outcomes between groups in all the analyses. Categorial variables between groups were compared using the chi-squared (X^2^) test. Because of the non-equivalence of the two groups in terms of sample size, we performed an additional analysis in which each patient who did not receive a TP (*n* = 45) was matched to a patient who received a TP (*n* = 45) for gender and age (±5 years). The same analyses were performed on this new sample to test the robustness of the results. Data were processed using the SPSS Statistics 24 (IBM Corporation, Armonk, NY, USA) software package. All results were considered statistically significant at the 5% critical level.

## 6. Results

### 6.1. Participants

From the 208 medical records of patients admitted to the “Château rouge” or “CNRF” during the years from 2014 to 2018 for multidisciplinary rehabilitation following a major amputation in the lower limbs, apparatus intent was noted for 171 of them. Among these 171 patients, some benefited from the TP technique (*n* = 126) and some were rehabilitated without this technique (*n* = 45). A flowchart to specify the sample size is available in Figure 2.

### 6.2. Descriptive Data

No differences were observed between groups in terms of age, gender, BMI, prevalence of vascular etiology of amputation, site of amputation, and time between amputation and entry to the rehabilitation center (all *p* > 0.05). The population characteristics are available in Table 1.

### 6.3. Outcome Data and Main Results

A summary of the impact of a TP on rehabilitation on discharge destination, length of stay, number of sockets required, and number of size adaptations of the first socket and functional level is available in Table 2.

After rehabilitation, 91.1% of the patients re-educated without using a TP returned home versus 82.5% of patients re-educated with TP. The other patients were oriented to nursing home facilities. No significant difference between the groups was observed (*p* = 0.17).

Regarding the length of stay in rehabilitation, a significantly longer average length of stay was observed for re-educated patients without a TP (median of 99 (80–154) days) when compared to those who used a TP during their rehabilitation (median of 68.5 (53–88) days) (*p* < 0.001).

The number of sockets used and the number of size adaptations of the first EP sockets were assessed across only 164 patients (43 patients and 44 stumps in the group of patients without a TP and 121 patients and 125 stumps in the group of patients with a TP). The maximum number of sockets per stump was two in both groups, and the maximum number of procedures of the size adaptation was three in the group of patients without TP and four in the group of patients with TP. The frequency of the number of sockets required for rehabilitated stumps was not significantly different between groups (*p* = 0.913), and the frequency of the number of interventions for the size adaptation of the first EP socket was not significantly different either (*p* = 0.597).

The functional group outcome was assessed in only 114 patients in the group of patients using TP (8 died before the DP could be prescribed and 4 were ultimately not fitted despite the initial intention) and 30 patients in the group of patients who did not use a TP (no data available for 15 patients). In the two groups, patients were most often classified in category 3 and least often classified in category 2. Category 1, which corresponds to patients benefiting from a prosthesis for aesthetic purposes only, is not included in this study because it does not correspond to patients for whom there was initially an intention to have a fitting. No significant difference was observed in regard to the distributions in these different categories (*p* = 0.83).

### 6.4. Other Analyses

The length of stay was significantly shorter for patients with TP. The influence of gender, age groups, BMI categories, and level of amputation on this outcome was studied via subgroup analyses (Table 3). A significantly lower length of stay in rehabilitation for patients with TP compared with patients without TP is only observed in men (*p* < 0.001) but not in women (*p* = 0.19). A similar observation is made for BMI categories where a significantly lower length of stay in rehabilitation for patients with TP compared with patients without TP is only observed in overweight/obese patients (*p* = 0.001) and not in normal/low BMI patients (*p* = 0.89). Moreover, a significantly lower length of stay in rehabilitation for patients with TP compared with patients without TP is only observed in patients with a tibial amputation (*p* < 0.001) and not in patients with femoral amputation (*p* = 0.22). No influence of age (<65 or ≥65 years) on the length of stay of patients with TP could be demonstrated.

We did not find additional associations between rehabilitation using TP and the other outcomes (discharge destination, number of sockets required, number of size adaptations of each socket, and functional level) in any of the subgroup analyses.

Additional analyses were performed on a sample of 90 participants, with 45 without a TP matched on age and 45 patients with a TP matched on gender (characteristics of the population are available in Appendix A). The post hoc paired analyses confirmed the findings, with a significantly lower length of stay for patients who used a TP compared to patients who did not (*p* < 0.001). Moreover, using these matched analyses, we also observed a higher number of patients that required two sockets for rehabilitation stumps in the group of patients that did not receive a TP compared to those that received it (*p* = 0.049) (data available in Appendix A).

## 7. Discussion

### Key Results

The objective of this study was to study the impact of integrating a training prosthesis in the rehabilitation of patients with lower limb amputation. This impact has been assessed through various outcomes (discharge destination, number of sockets and size adaptations of the first socket, length of stay, and functional group). TP exhibits a positive impact on the length of stay, whereas there is no relation with the other investigated outcomes. This is an original article and the first of its kind; therefore, the absence of prior studies prevents comparisons with the literature.

## 8. Interpretation

One of the outcomes studied was the discharge destination. No difference was highlighted between the two groups for this point. Regarding patients with lower limb amputation, factors influencing the discharge destination have been mentioned in the literature, such as age, amputation level, and comorbidities [21,22,23]. However, clinical features did not appear to be associated with a particular destination [23].

Concerning the results that relate to the number of sockets or size adaptations of sockets, it is tempting to imagine that the results described in the literature on IPOP can be extrapolated to those of TP. IPOP showed an impact on the scarring, shape, and edema of the stumps with assessments over the short term [14]. Even if this has not been directly studied, one could therefore imagine a decrease in the number of sockets or socket adaptations in patients who have benefited from an IPOP compared to other amputee patients. However, IPOP and TP are not used in the same phase. They come into play, respectively, in the acute post-surgical phase and the pre-prosthetic phase of lower limb amputation. We therefore chose to assess the stability of the stumps by focusing on the sockets of the EP. Thus, the appearance of the stump is evaluated after the amputation than what is generally proposed in the literature. However, the volume of the stump takes several months to stabilize [24], although there are significant variations between individuals, which do not allow clinicians to determine a precise time when the volume is stable [25]. It has been suggested that the use of temporary prostheses results in the fastest reduction in stump volume when compared to elastic bandages or pneumatic prosthetics [26]. We could thus have expected a lower number of socket size adaptations in the group of patients re-educated with TP. Our study was unable to demonstrate such a relationship between the use of a PT and the number of sockets or size adaptation of the prosthesis. This outcome is, however, an imprecise tool for studying the volume of the stump. Other precise methods of stump measurement have been proposed [27]. Unfortunately, we could not use these methods in our study because of the retrospective design and were therefore unable to assess them.

Regarding the length of stay in rehabilitation centers, a very clear difference is observed, with a significantly shorter stay for patients whose rehabilitation included the use of TP (68.5 versus 99 mean days). This could be explained by the faster acquisition of autonomy, by the earlier healing of the stump, or perhaps even by faster reintegration into socio-family life. This question deserves to be studied more specifically by evaluating the stump’s healing time, quality of life questionnaires, and an evaluation of the patient’s autonomy: for example, by measuring functional independence. One can imagine the impact of such a reduction in the length of stay at the economic level by multiplying the cost of a day of hospitalization by the difference between the two average lengths of stay. Similarly, this reduction in the length of stay has a direct impact on the reduction in morbidity linked to prolonged hospitalizations, as well as the socio-family sphere. Shortening the length of hospital stay also improves patient flow in the context of increasing demand for hospital services [28].

The impact of the use of TP on the reduction in length of stay seems particularly important in men, patients with a BMI ≥ 25 kg/m^2^, and patients with tibial amputation. Regarding the BMI, we can imagine that a larger volume and a fattier composition of the stump would require more time and more work to attain a sufficiently stable situation. Therefore, the TP plays a more important role in this process in patients with a greater BMI. In addition, a study showed that a lower functional independence measure (FIM) upon admission was associated with a longer length of stay [29]. It could be suggested that factors responsible for poorer mobility, such as a higher BMI, would also be associated with a longer length of stay. Thus, due to the early mobility that the training prosthesis allows, it could be explained that the reduction in the length of stay is especially marked for less mobile patients with a higher BMI. The same study also showed a longer length of stay for patients with femoral rather than tibial amputation. However, in our study, the TP was particularly effective in reducing the length of stay of patients with tibial amputation. We can imagine that, in this case, the early gain in mobility provided by the training prosthesis is not counterbalanced by the time required to learn to walk with the control of a prosthetic knee. Therefore, the effect of PT on the reduction in length of stay would be particularly beneficial to tibial amputees who do not need to learn to use a prosthetic knee. We do not have a specific hypothesis that explains the larger impact of TP on men’s length of stay, and we believe that these observations should be verified through other studies before confirming them.

Finally, the functional group reached by the patient when the final prosthesis was prescribed was analyzed and compared between the two groups. It would have been imaginable to observe better functionality in patients rehabilitated with TP. However, no difference was observed between re-educated patients with TP and re-educated patients without TP. This could be explained by the fact that the functionality in this study is understood by the functional group reached by the patient only at the time of prescribing DP, which is after at least six months of using EP. After such a delay, it is understandable that the functionality of the two groups is comparable. A significant difference in the functional capacities of amputee patients could perhaps have been found at an earlier phase of rehabilitation.

### 8.1. Generalizability

The pre-prosthetic phase of rehabilitation has not been studied extensively in the literature. This study, which covers 171 patients, made it possible to evaluate the techniques of training prostheses. It demonstrated that the use of a training prosthesis seems to be of interest in improving the rehabilitation of patients with lower limb loss by shortening the length of stay. This innovative technique has been poorly studied in the literature so far, and additional studies using preferentially prospective designs are required before confirming these results. The use of the training prosthesis could also be combined with other rehabilitation techniques, such as attention rehabilitation, which has exhibited good efficacy in physical performance and patients with musculoskeletal disorders [30,31].

### 8.2. Limitations

There are some limitations in this study. First, no validated scale was used to assess rehabilitation outcomes. A recent study confirmed that there is a lack of standardized sets of outcome measurements for patients with amputation throughout the literature [32]. This should be developed in order to conduct studies that are more robust, and subsequently, meta-analysis should be used.

The main limitations of this study are inherent in its retrospective nature. It would be ideal to be able to carry out the same work but in a prospective manner, which would provide the possibility on the one hand to integrate other observation tools—for example, quality of life questionnaires or functional scales—carried out at the different centers simultaneously with rehabilitation and, on the other hand, decrease the number of unknown data. This would also allow patients to be randomized into re-educated groups with or without TP. Unfortunately, few centers have the technical infrastructure and personal resources to carry out the TP. In this study, CHU-CNRF regularly integrates the training prosthesis technique into the rehabilitation of amputee patients, while CHR-Château Rouge does not use it. It would therefore be necessary to conduct the study only in a center that offers this technique, which would reduce the sample or increase the duration of the study, or develop the TP technique at an additional center. Given the results of this study, it would be very interesting to encourage the use of TP in other centers and conduct new research on the pre-prosthetic phase of rehabilitation.

## Figures and Tables

**Figure 1 healthcare-12-00567-f001:**
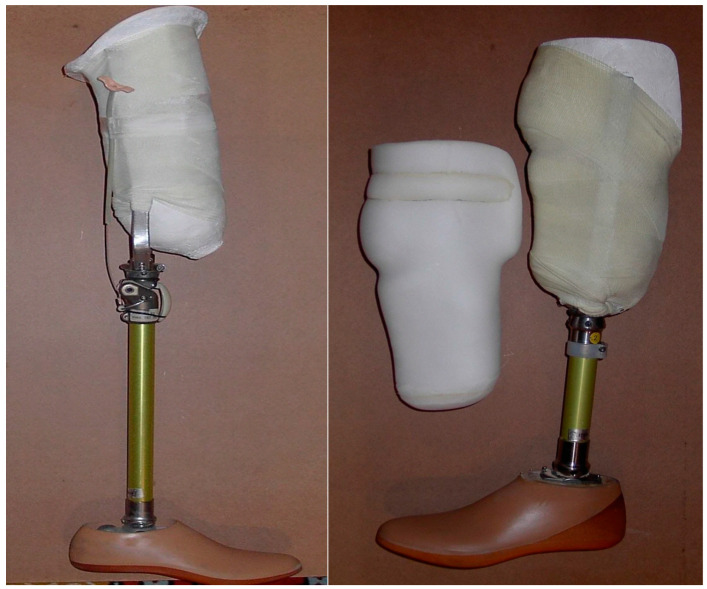
Femoral and tibial training prosthesis.

**Figure 2 healthcare-12-00567-f002:**
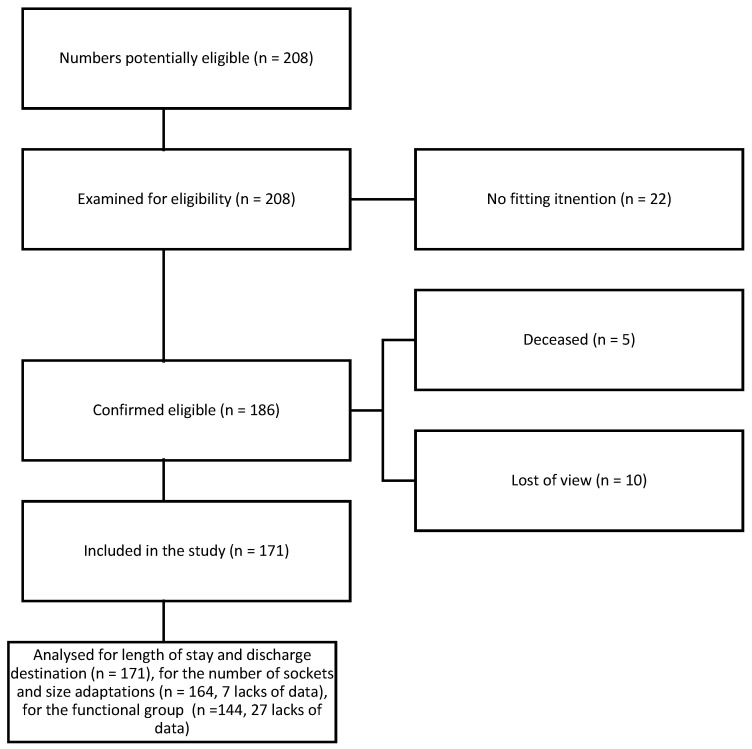
Sample size flow chart.

**Table 1 healthcare-12-00567-t001:** Population characteristics.

	Rehabilitation with TP (n = 126)	Rehabilitation without TP (n = 45)	*p*-Value ^a^
Gender (men)	93 (73.8)	32 (71.1)	0.73
Age (years)	66.5 (57.7–74)	62 (55.5–70.5)	0.47
BMI (kg/m^2^)	23.2 (20.1–27.3)	26.5 (20.9–31.4)	0.18
Vascular etiology (yes)	110 (87.3)	40 (88.4)	0.78
Site of amputation Unilateral above-knee amputation (yes) Unilateral below-knee amputation (yes) Bilateral (yes)	54 (42.9) 68 (54.0) 4 (3.2)	16 (35.6) 27 (60.0) 2 (4.4)	0.67
Time between amputation and entrance into the rehabilitation center (days)	20 (14–28)	16 (12–27)	0.10

TP: training prosthesis; BMI: body mass index. Qualitative variables are expressed in absolute (n) and relative (%) frequencies; quantitative variables are expressed in the median and interquartile range (P25–P75). ^a^
*p*-values obtained from the Mann–Whitney U test for continuous variables and the X^2^ test for categorical variables.

**Table 2 healthcare-12-00567-t002:** Outcomes associated with rehabilitation with TP.

	n	Rehabilitation with TP	Rehabilitation without TP	*p*-Value ^a^
Discharge destination Back to home To nursing facilities	171	104 (82.5) 22 (17.5)	41 (91.1) 4 (8.9)	0.17
Length of stay in rehabilitation center (*days*)	169	68.5 (53–88)	99 (80–154)	<0.001
Number of sockets required (*nbr*) 1 socket 2 sockets Interventions for first EP socket (*nbr*) None 1 size adaptation 2 size adaptations ≥3 size adaptations	164 164	107 (88.4) 14 (11.6) 58 (47.9) 32 (26.4) 20 (16.5) 11 (9.1)	37 (86.0) 6 (13.3) 24 (55.8) 9 (20.9) 7 (16.3) 3 (7.0)	0.68 0.83
Functional group Group 0 Group 2 Group 3 Group 4 Group 5	144	25 (19.8) 2 (1.6) 52 (41.3) 19 (15.1) 16 (12.7)	4 (13.3) 1 (0.33) 16 (53.3) 5 (16.7) 4 (13.3)	0.83

TP: training prosthesis; EP: exercise prosthesis. Qualitative variables are expressed in absolute (n) and relative (%) frequencies; quantitative variables are expressed in the median and interquartile range (P25–P75). ^a^
*p*-values obtained from the Mann–Whitney U test for continuous variables and the X^2^ test for categorical variables.

**Table 3 healthcare-12-00567-t003:** Median length of stay (days) in the rehabilitation center for the group with TP and group without TP according to subgroups.

	n	Rehabilitation with TP Median (P25–P75)	Rehabilitation without TP Median (P25–P75)	*p*-Value ^a^
Gender				
Men	124	66 (52–85)	101 (80–164)	<0.001
Women	45	77 (64–92.5)	93 (67–138.2)	0.76
Age				
<65 years	83	60 (46.5–85.5)	99 (63–153.2)	0.005
≥65 years	86	76 (61.5–92)	94 (84–159)	0.005
BMI				
<25 kg/m^2^	91	70 (57–86.2)	67 (48.5–89.5)	0.89
≥25 kg/m^2^	67	66 (51–94.5)	99 (87–148)	0.001
Level of amputation				
Femoral	69	68 (57–95)	92 (46–101)	0.22
Tibial	94	71 (51.2–84.7)	105 (83.7–156.5)	<0.001

TP: training prosthesis; BMI: body mass index. Quantitative variables are expressed in the median and interquartile range (P25–P75). ^a^
*p*-values obtained from the Mann–Whitney U test.

## Data Availability

Data are available upon reasonable request to the corresponding author.

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
