# Peer review of "Outcomes of Patients with Lower Limb Loss after Using a Training Prosthesis: A Retrospective Case Series Study"

_healthcare, 2024, doi:10.3390/healthcare12050567_

Round 1
Reviewer 1 Report
Comments and Suggestions for Authors
Dear Authors,
It is my pleasure to review your study. The study is interesting but I have a lot of doubts.
General information:
-There are no clear conclusions in the abstract. It should be corrected.
-References should be prepared according to the journal's guidelines.
-Some references are incomplete. This needs to be completed.
-Moreover, the references are relatively old. It would be good to add some newer references.
-References in the text should be placed in square brackets.
Introduction:
-The introduction is written correctly, I have no objections.
M&M:
-Inclusion and exclusion criteria are unclear and illegible. It should be corrected.
-"However, the CHU-CNRF regularly integrates the training prosthesis technique into the rehabilitation of amputee patients, while the CHR-Château Rouge does not use it." It should be added to the limitations section.
-"2.3. Outcomes investigated in Functional group." Please add reference to this informations about five categories.
- This is retrospective study. Has sample size been counted? No information about it in the manuscript. This is very important.
-Why was it adopted: "126 benefited from the TP technique and 45 were rehabilitated without this technique" and not e.g. n =45 and n=45? This needs to be explained.
Results:
-What does it mean "Functional group, group 0-5" ?
-What was the rehabilitation program? This has not been described. Please explain it.
-Were any scales used to assess effectiveness?
Some patients died. Some data are missing. In retrospective studies, sample size should be calculated. And perform statistical analysis on the entire group. Requires serious analysis and correction.
Discussion:
-Newer references should be added.
-Limitations section should be refined.
First, methodological issues related to the study design must be resolved.
Comments on the Quality of English LanguageEnglish correction required.
Author Response
Reviewer #1
General information:
-There are no clear conclusions in the abstract. It should be corrected.
Answer: First of all, we want to thank you for the time you took to review our work and propose some very relevant and constructive changes. Thank you for your suggestion about the conclusion in the abstract. We corrected this in the abstract.
-References should be prepared according to the journal's guidelines.
Answer: Thank you for noticing. We modified the presentation of the references in accordance with the guidelines.
-Some references are incomplete. This needs to be completed.
Answer: Thank you for noticing. We completed the incomplete references.
-Moreover, the references are relatively old. It would be good to add some newer references.
Answer: Thank you for the suggestion. We added new interesting references, with a high level of evidence.
-References in the text should be placed in square brackets.
Answer: Thank you for noticing. We changed this presentation throughout the manuscript.
Introduction:
-The introduction is written correctly, I have no objections.
Answer: thank you for your comment.
M&M:
-Inclusion and exclusion criteria are unclear and illegible. It should be corrected.
Answer: thank you for this suggestion. We made the presentation of inclusion and exclusion criteria much clearer and simpler.
-"However, the CHU-CNRF regularly integrates the training prosthesis technique into the rehabilitation of amputee patients, while the CHR-Château Rouge does not use it." It should be added to the limitations section.
Answer: thank you for this suggestion. We added this precision in the limitations section, as you recommend it.
-"2.3. Outcomes investigated in Functional group." Please add reference to this information about five categories.
Answer: thank you for the suggestion. We added this sentence in the concerned paragraph: “(…)in accordance with the Belgian health insurance https://www.inami.fgov.be, also available in appendix”. We also added the material in appendix, for ease of reference for the reader
- This is retrospective study. Has sample size been counted? No information about it in the manuscript. This is very important.
Answer: thank you for your question. We added detailed information about sample size in our manuscript, according to STROBE recommendations and presented the flow chart in figure 2. for clear of use.
-Why was it adopted: "126 benefited from the TP technique and 45 were rehabilitated without this technique" and not e.g. n =45 and n=45? This needs to be explained.
Answer: thank you for your comment. For ease of use, this has been changed as follow: “Among these 171 patients, some benefited from the TP technique (n = 126) and some were rehabilitated without this technique (n = 45)”
Results:
-What does it mean "Functional group, group 0-5" ?
Answer: for more comprehension, this information has been added in the appendix.
-What was the rehabilitation program? This has not been described. Please explain it.
Answer: Thank you for the suggestion. It has been added to the M&M section (population) as follow: “All patients received physiotherapy and occupational therapy to work on strengthening, gait rehabilitation, balance and functionality in activities of daily living. When necessary, they also received speech therapy, psychology and neuropsychology care.”
-Were any scales used to assess effectiveness?
Answer: Unfortunately, no scale has been used consistently to assess the effectiveness of training prostheses. This is why we have started another study, with a prospective design, in order to be able to integrate validated and reproducible rating scales for each patient, at the same moment of the rehabilitation.
- Some patients died. Some data are missing. In retrospective studies, sample size should be calculated. And perform statistical analysis on the entire group. Requires serious analysis and correction.
Answer: thank you for this comment. We added a sample size flow chart and we explained the groups on which we realized the statistical analysis (figure 2). We also adapted our manuscript according to the STROBE recommendations.
Discussion:
-Newer references should be added.
Answer: thank you for your comment. We added new references in the discussion, in connection with other of your comments. For example: “There is some limitations of this study. First of all, no validated scale was used to assess rehabilitation outcomes. A recent study confirmed that there is a lack of standardized sets of outcomes measurements for patients with amputation throughout the literature”. This should be developed, in order to conduct studies more robust and subsequently meta-analysis.” And “In addition, a study showed that a lower functional independence measure (FIM) at admission was associated with a longer length of stay”.
-Limitations section should be refined.
Answer: thank you for your suggestion. We reorganized the “limitations” section to make it clearer.
First, methodological issues related to the study design must be resolved.
Answer: thank you for your relevant comment. We reworked our manuscript according to the STROBE recommendations.
Reviewer 2 Report
Comments and Suggestions for Authors
The effectiveness of training prostheses is verified using data from clinical practice. In addition to the large number of patients, this paper not only verifies that training prostheses are effective, but also explores for which subgroup the training prostheses are more effective. I think it is a valuable research paper.
I have no particular concerns, but I think the discussion chapter was a bit lacking, especially in lines 274-294. These paragraphs should mention a more in-depth discussion of why training prostheses were effective and whether they were particularly effective for certain subgroups. Also, it would be better to provide more detailed information and pictures of the training prostheses.
Author Response
Reviewer #2
The effectiveness of training prostheses is verified using data from clinical practice. In addition to the large number of patients, this paper not only verifies that training prostheses are effective, but also explores for which subgroup the training prostheses are more effective. I think it is a valuable research paper.
I have no particular concerns, but I think the discussion chapter was a bit lacking, especially in lines 274-294. These paragraphs should mention a more in-depth discussion of why training prostheses were effective and whether they were particularly effective for certain subgroups. Also, it would be better to provide more detailed information and pictures of the training prostheses.
Answer: Thank you very much for your time, your comment and yours suggestions. Pictures of a tibial and of a femoral training prosthesis are now available in Figure 1. We have also expanded the discussion with new references, including detailing the possible impact of the training prosthesis on the different subgroups, as suggested.
Reviewer 3 Report
Comments and Suggestions for Authors
Thanks a lot for the opportunity you have offered me to revise the fascinating manuscript " Outcomes of patients with lower limb loss after using a training prosthesis: A retrospective case series study".
As a significant strength, this manuscript investigate outcomes in patients with lower limb loss, based on whether or not they used a training prosthesis (TP) during their rehabilitation. This proposal is a novelty in the field and adds information to the existing evidence in the literature produced in the field.
As a major weakness, the manuscript sometimes needs more details and clarity concerning methodological steps that would help improve the understanding of the manuscript. Therefore, I have suggested some strategies to improve authors' reporting and increase the quality of their work.
Overall, my peer review is a minor revision.
¶MAJOR ISSUES:
#INTRODUCTION:
*rationale: the rationale is quite good. I suggest authors emphasize the novelty of their study, showing the existing evidence on the topic presented in literature.
*main questions: I suggest authors explain the rationale behind their question better. Moreover, I would like to know why a retrospective study was chosen instead of a prospective design. What is its value?
#METHODS:
*STROBE: The study is a retrospective case-control study. Although it is well structured, I suggest the authors organise it according to STROBE (https://www.strobe-statement.org) and make it explicit in their materials and methods.
*bias: how did authors control methodological bias in this retrospective study?
*inclusion and exclusion criteria should be based on evidence. Please, do it.
*rehabilitation: Is it possible to have more details about the rehabilitation path that patients received?
#DISCUSSION:
*main findings: what are the main findings of the study? Authors should explicitly show them. It is important to show the element of novelty of their study compared to the other published evidence.
*Implications for clinical practice and resarch: I suggest authors explain well the implications for clinicians and researchers. For example, functional exercises that use an external focus of attention could be really good for this category of patients (doi: 10.3390/jfmk3030040 doi: 10.1038/s41598-018-30228-9). Moreover, future studies should investigate EMG activity and inter-limb asymmetry during walking (doi: 10.1038/s41598-022-07975-x) and analyse the body-impaired perceptions (doi: 10.3389/fnhum.2020.00083) presented in patients with prostheses. The authors should read and integrate these references on the topic.
¶MINOR ISSUES:
#ENGLISH:
*need for manuscript revision: I suggest the authors have the English revised by a native speaker. In fact, there are several typos and inaccuracies (e.g., i.e.).
#REFERENCES:
*the references seem to be dated. I suggest authors refer to recent findings with a high levels of evidence (e.g., systematic review, metanalysis). This strategy will improve the quality of your paper.
#TABLES
*Acronyms. The tables are good. However, I suggest reporting in full all the acronyms presented in the whole manuscript (e.g., in the Abstract, there is BMI).
Comments on the Quality of English LanguageMinor English revision is needed.
Author Response
Reviewer #3
Thanks a lot for the opportunity you have offered me to revise the fascinating manuscript "Outcomes of patients with lower limb loss after using a training prosthesis: A retrospective case series study".
As a significant strength, this manuscript investigate outcomes in patients with lower limb loss, based on whether or not they used a training prosthesis (TP) during their rehabilitation. This proposal is a novelty in the field and adds information to the existing evidence in the literature produced in the field.
As a major weakness, the manuscript sometimes needs more details and clarity concerning methodological steps that would help improve the understanding of the manuscript. Therefore, I have suggested some strategies to improve authors' reporting and increase the quality of their work.
Overall, my peer review is a minor revision.
¶MAJOR ISSUES:
#INTRODUCTION:
*rationale: the rationale is quite good. I suggest authors emphasize the novelty of their study, showing the existing evidence on the topic presented in literature.
Answer: thank you for your comment and your suggestion. We now emphasized the novelty of our study as: “Although its use is not uncommon, there is a lack of studies on this topic in literature. However, its use could have an impact on several key rehabilitation outcomes(19), such as length of stay, discharge destination, quality of life, functionality, economic burden, prosthesis fitting, etc.
The aim of our work was therefore to investigate different health rehabilitation outcomes after rehabilitation of lower limb loss patients, based on whether the patients used a training prosthesis in the rehabilitation program.”
*main questions: I suggest authors explain the rationale behind their question better. Moreover, I would like to know why a retrospective study was chosen instead of a prospective design. What is its value?
Answer: thank you for your comment. In fact, no studies of any type of design were available (at least in the English-language literature) about the impact of training prostheses on rehabilitation outcomes. For this reason, we performed a retrospective study, which is presented in this article. We also want to carry out a similar study with a prospective design. However, in order to be as representative as possible of the general population and to increase the sample size, we want this study to be multicenter, and this requires the formation of teams in the different centers included as well as the installation of the necessary infrastructure for the manufacture of training prostheses. Funding efforts are underway. We look forward to seeing if the results of this prospective study will match the results of our retrospective study.
#METHODS:
*STROBE: The study is a retrospective case-control study. Although it is well structured, I suggest the authors organise it according to STROBE (https://www.strobe-statement.org) and make it explicit in their materials and methods.
Answer: Thank you for your recommendation. We restructured our article according to STROBE, made this clear in materials and methods section.
*bias: how did authors control methodological bias in this retrospective study?
Answer: thank you for your comment. We added a bias section, according to STROBE recommendation “The sample size wasn’t equivalent between the two groups. In order to test the robustness of the results, we realized the same analyze on a new sample, composed with the 45 patients that did not receive a TP, with 45 matched patients that received a TP. There were matched for gender and age (±5years).”
*inclusion and exclusion criteria should be based on evidence. Please, do it.
Answer: thank you for your recommendation. We added it in the corresponding section: “Patients were included in the study if they had undergone a major unilateral or bilateral amputation of the lower limb (above the ankle and below the hip), without restriction of age, comorbidity or etiology. Major amputations were selected because they show a difference compared to minor amputations in terms of rehabilitation outcome (walking ability, quality of life, dependency status)”.
*rehabilitation: Is it possible to have more details about the rehabilitation path that patients received?
Answer: thank you for your suggestion, we added a paragraph “All patients received physiotherapy and occupational therapy to work on strengthening, gait rehabilitation, balance and functionality in activities of daily living. When necessary, they also received speech therapy, psychology and neuropsychology care”.
#DISCUSSION:
*main findings: what are the main findings of the study? Authors should explicitly show them. It is important to show the element of novelty of their study compared to the other published evidence.
Answer: thank you for your recommendation. We have now dedicated the key results section to this aspect: “The objective of this study was to study the impact of integrating a training prosthesis in the rehabilitation of patients with lower limb amputation. This impact has been assessed through various outcomes (discharge destination, number of sockets and size adaptations of the first socket, length of stay, functional group). TP shows to have a positive impact on length of stay, whereas there is no relation with other outcomes investigated. This is an original article, the first of its kind, and therefore the absence of prior studies prevents comparisons with literature”.
*Implications for clinical practice and resarch: I suggest authors explain well the implications for clinicians and researchers. For example, functional exercises that use an external focus of attention could be really good for this category of patients (doi: 10.3390/jfmk3030040 doi: 10.1038/s41598-018-30228-9). Moreover, future studies should investigate EMG activity and inter-limb asymmetry during walking (doi: 10.1038/s41598-022-07975-x) and analyse the body-impaired perceptions (doi: 10.3389/fnhum.2020.00083) presented in patients with prostheses. The authors should read and integrate these references on the topic.
Answer: thank you for your suggestion. We added new lines about such clinical research and practice: “The use of the training prosthesis could also be combined with other rehabilitation techniques, such as focus of attention, which has shown good efficacy in physical performance and in patients with musculoskeletal disorders”. We didn’t integrate references about EMG investigations during walking since few muscles are bilaterally accessible for amputee patients and it is generally preferred to use techniques such as inertial sensors for the assessment of asymmetry in gait for this category of patients.
¶MINOR ISSUES:
#ENGLISH:
*need for manuscript revision: I suggest the authors have the English revised by a native speaker. In fact, there are several typos and inaccuracies (e.g., i.e.).
Answer: thank you for your suggestion. We’ll plan a language revision.
#REFERENCES:
*the references seem to be dated. I suggest authors refer to recent findings with a high levels of evidence (e.g., systematic review, metanalysis). This strategy will improve the quality of your paper.
Answer: thank you for your recommendation. We added news references with high levels of evidence.
#TABLES
*Acronyms. The tables are good. However, I suggest reporting in full all the acronyms presented in the whole manuscript (e.g., in the Abstract, there is BMI).
Answer: thank you for this suggestion. We corrected the acronyms.
Round 2
Reviewer 1 Report
Comments and Suggestions for Authors
Dear Authors,
Thank you for revising the manuscript. The article looks much better.
In my opinion, there are still some minor changes that need to be made:
1. In line: 102 "All patients received physiotherapy and occupational therapy to work on strength- 102 strengthening, gait rehabilitation, balance, and functionality in daily living activities. When nec- 103 essary, they also received speech therapy, psychology, and neuropsychology care."
Please provide time spent on rehabilitation.
2. The limitations section should be at the end of the discussion.
Apart from that, I have no comments.
Best regards,
Author Response
Dear Editor,
We would like to thank again the reviewers for their time spent on our manuscript. We fully agree with the suggestions. Please find below point by point our response to the reviewers’ comments. We are hopeful that this is now suitable for publication.
Dr Doriane Pelzer
Point par point rebuttal letter :
Reviewer #1
Dear Authors,
Thank you for revising the manuscript. The article looks much better.
In my opinion, there are still some minor changes that need to be made:
- In line: 102 "All patients received physiotherapy and occupational therapy to work on strength- 102 strengthening, gait rehabilitation, balance, and functionality in daily living activities. When nec- 103 essary, they also received speech therapy, psychology, and neuropsychology care."
Please provide time spent on rehabilitation.
Answer: thank you for your suggestion. We changed the paragraph as follows:
“All patients received 2 hours of rehabilitation each day. The treatment included physiotherapy and occupational therapy to work on strengthening, gait rehabilitation, balance, and functionality in daily living activities. When necessary, they also received speech therapy, psychology, and neuropsychology care.”
- The limitations section should be at the end of the discussion.
Apart from that, I have no comments.
Best regards,
Answer: thank you for the suggestion. We reorganized the discussion section as suggested.